# Evaluation of the Economic and Environmental Sustainability of Livestock Farms in Inland Areas

Michele Cerrato, Allegra Iasi, Federica Di Bennardo and Maria Pergola *

Degree Course of Agriculture, Dipartimento di Farmacia, Università degli Studi di Salerno, Via Giovanni Paolo II 132, 84084 Fisciano, Italy; mcerrato@unisa.it (M.C.); a.iasi@studenti.unisa.it (A.I.); dibennardofederica@gmail.com (F.D.B.)
* Correspondence: mpergola@unisa.it; Tel.: +39-320-7875-447

**Abstract:** The present research aimed to evaluate the economic and environmental sustainability of livestock farms in inland areas of the Cilento, Vallo di Diano and Alburni National Park (Southern Italy) and the convenience and possibility of activating forms of local economies. The study involved three types of grazing husbandries: one with only sheep and goats; one with only cattle; and one mixed, namely with cattle, sheep and goats. The profitability of the analyzed farms was compared through their gross profit and the net income of the farmer. To evaluate the convenience of the farms under study to activate forms of a short supply chain, the transformation value of the milk was used as the reference parameter. The environmental impact *per* farm and *per* adult bovine unit was assessed through the LCA methodology. The economic analysis showed that the survival of the analyzed farms is essentially linked to public subsidies, which in some cases represent more than 75% of the total output. Family enterprise plays a fundamental role in management decisions, in the size of animal breeding, and in investment decisions. Referring to environmental impacts, the analysis showed a lower sustainability of cattle farming, mainly due to the higher methane emissions during enteric fermentation. Despite all this, the ecosystem services provided by these semi-extensive farms in inland areas are significant, and therefore economic and environmental analyses should take them into account to enhance them and encourage farmers to remain in these often marginal areas.

**Keywords:** rural development; life cycle assessment; profitability; livelihood; rural household; animal husbandry



## 1. Introduction

The Cilento, Vallo di Diano and Alburni National Park (Southern Italy), declared a World Heritage Site by UNESCO in 1998, is very rich in biodiversity, as evidenced by the presence of numerous Sites of Community Importance (SIC) [1]. Here, animal husbandry is of considerable importance both from an environmental point of view, through the functions of fire prevention, forest cleaning, and soil erosion control, and from an entrepreneurial and economic point of view, in support of families for food production [2].

In the Campania region, there are about 18,400 breeding farms, mainly with cattle (about 9580), followed by flocks of sheep (about 5000) and goats (2573) [3]. The ecosystem services (ES) they provide are numerous, especially those provided by small breeders and grazing animals. The concept of ES has been extensively developed and gained wide acceptance through the Millennium Ecosystem Assessment (MEA) [4]. The MEA distinguished four groups of ecosystem services: (1) provisioning services, referring to products obtained from ecosystems (food, fiber, fertilizer, fuel, etc.); (2) regulating services, referring to the benefits obtained from the regulation of ecosystem processes (waste recycling, land degradation and erosion prevention, etc.); (3) supporting services, necessary for the production of all other ecosystem services (maintenance of soil structure and fertility, maintenance of genetic diversity, etc.); and (4) cultural services, which refer to the non-material benefits people obtain from ecosystems through spiritual enrichment, cognitive development,

reflection, recreation, and aesthetic experiences [5]. A second key initiative on ES is the Economics of Ecosystems and Biodiversity [6], which defined ES as "the direct and indirect contributions of ecosystems to human well-being", separating services from benefits to explicitly identify the services that provide multiple and indirect benefits [5].

Livestock and breeders are key components of agro-ecosystems and interact closely with natural ecosystems and therefore play an essential role in the provision of ES [5]. Livestock's interaction with other ecosystem components and processes is more complex than that of plants, because of livestock's higher position in the trophic network, which results in conversion losses and associated environmental externalities. In particular, there are three characteristics of livestock that shape their specific roles in ecosystems: (1) livestock's unique ability to convert non-human-edible feed and organic waste into useful products, through their digestive tracts; (2) the direct nature of thtrophic networkeir interactions with ecosystems (e.g., land, vegetation and soil) through trampling, grazing and browsing, as well as the production of urine and dung; and (3) their mobility and resulting ability to respond to temporal and spatial fluctuations of ecosystems in resource availability. Moreover, the contribution of livestock species and breeds to ES is intimately tied to the production systems they are associated with and hence the diverse human management systems affecting them [5].

At the same time, livestock production causes more than 80% of agriculture's greenhouse gas emissions (GHG) and uses about 70% of total agricultural land in the European Union (EU) [7]. Ruminants are among the livestock sector's major contributors to global warming, generating emissions from enteric fermentation, feed production, manure management, energy consumption in barns and deforestation [8,9].

One of the most commonly used methodologies to estimate environmental impacts is life cycle assessment (LCA), a cradle-to-grave methodology to assess products, processes, services, activities and systems based on the life cycle thinking approach [10]. LCA has been proven to be a valuable tool for addressing questions about the environmental impact of various agricultural production systems [11], resting on both the identification of the subsystems that contribute most to the total environmental impact and the comparison of products and processes with the same functions [12–18]. In the literature, there are several LCA studies involving the zootechnical sector. The most recent and interesting concern: the assessment of the environmental impact of grazing farms in the Republic of Ireland [19]; a localized agricultural LCA database to calculate GHG emissions, which made the first extensive assessment of smallholder farms' GHG emission reduction potential by coupling crop and livestock production [20]; a review of methodologies published to date that combine animal welfare evaluation with LCA [21]; a carbon footprint assessment, combining various scales of analysis and including a territorial assessment, to estimate the GHG emissions from crops and livestock in an Indian village impacted by both the Green (for crops) and White (for milk) revolutions [22]. There are also numerous published studies focusing specifically on cattle [8,23], pigs [24], sheep [25], and poultry [26]. However, to our knowledge, there are no recent LCA studies on animal husbandry evaluating together the environmental impact and the profitability of livestock farms, especially in inland mountain areas.

In light of what has been said so far, the aims of the present research were to evaluate the economic and environmental sustainability of representative livestock farms in a rural area supported and enhanced by the Local Action Group (LAG) "Casacastra" (Salerno province) and the possibility of activating forms of local economies through the transformation of milk on the farm.

## 2. Materials and Methods

### 2.1. Data Collection

The study was carried out in the Campania region (Italy), precisely in Centola and Casaletto Spartano municipalities (Salerno province), and involved three types of husbandries notably widespread in the Cilento area [2]: one with only sheep and goats; one with

only cattle; and one mixed, namely with cattle, sheep and goats. The three analyzed farms were selected without any statistical criteria, and therefore the limits of representativeness are considerable. In selecting the farms, the only criteria adopted were the location in the LAG "Casacastra" (Cilento area) and the similarity with the typologies of the farms spread in the study area. Thus, based on the willingness of the farmers to collaborate in carrying out the study, a farm only with ovines and caprines (OC FARM), one with only bovines (B FARM) and one mixed (M FARM) were identified.

The survey was conducted in 2022, and the collection of data useful for the economic and environmental analysis was carried out using a specially structured questionnaire. The questionnaires were compiled by expert and specially trained personnel interfacing with farm personnel suitable for the compilation. Where necessary, the data reported in the questionnaire were integrated with information derived from the farm accounts. The collected data were processed using an Excel spreadsheet and subsequently structured into tables and graphs.

As regards the calculation of the transformation value of the milk, data and information necessary for the analysis were collected in a cheese factory willing to collaborate in this research and which transformed the same average quantity of milk produced daily by the farms analyzed. Also in this case, data collection was carried out using a questionnaire prepared for the occasion and filled in by personnel suitable for the purpose of the research.

*2.2. Economic Analysis*

The objective of the economic analysis was to compare the profitability of the three farms analyzed. For this purpose, information (yields, labor and material inputs, use of fixed capital) was collected from the analyzed farms throughout 2022 and then converted into economic information, imputing prices and tariffs recorded on the local marketplace in the 2022/2023 agricultural year [27]. Economic results were expressed at constant values, and the profitability of the analyzed farms was compared through: (1) the gross profit (GP), obtained by deducting from the revenue (sales of products: total output—TO), all variable production costs (VC), gross of taxes and overheads; and (2) the net profit (NP) of the farmer, obtained by deducting from the TO all the production costs (variable and fixed costs: VC + FC = total cost). Analyses and comparisons were made both by the farm and by the adult bovine unit (ABU).

The asset of the balance sheet was calculated by adding the products sold by the farms under study (sheep and goats, stabled calves, adult cattle); gross barn profit (obtained by adding the increase in value of livestock during 2022 to sales and subtracting the value of any purchases); contributions to production, i.e., the compensation provided by the Common Agricultural Policy (CAP) per hectare, in favor of cereal producers and per head reared. Due to a lack of information, products (animals, milk and cheese) intended for self-consumption and gifts were not considered.

Referring to liabilities, for each farm, only explicit costs (actual cash outflows) were considered, and the following items were accounted for in determining them:

- Sundry expenses: the amount of this item was calculated by multiplying the quantities purchased by the corresponding market prices in force in the areas under investigation. Specifically, these expenses included those for cultivation (costs for seeds, fuel and lubricants); those for animal breeding, i.e., purchases of medicinal and sanitary products, as well as professional services and consultancy; and expenses for off-farm feeding (feed, hay, straw, by-products, supplements, etc.);
- Quotas: represented the charges for the partial reintegration of capital assets. This item concerned those assets whose use was carried out in several production cycles and specifically concerned depreciation, maintenance and insurance. The annual depreciation rates have been calculated for both buildings and machinery and equipment. The calculation was made using the financial depreciation criterion to also consider the related interest. For machinery and equipment, the annual depreciation rate was obtained by subtracting the recovery value from the purchase value. The difference

thus obtained was multiplied by a normalization coefficient ($rq^n/q^{n-1}$) [28]. In the case of buildings, the annual fee was calculated by multiplying the same normalization coefficient by the current reconstruction value. The maintenance and insurance quotas were obtained by applying a percentage (4%) to the purchase value of the machines and equipment, as well as to the reconstruction value of the buildings (2%). The cattle reintegration quota has not been calculated as the presence within the farms of the categories destined for replacement was sufficient to ensure a gradual replacement of breeding cattle;

- Labor: the cost of labor was not considered since it was only an implicit cost. In fact, as will be better specified in the descriptive part of the individual farms, the work was provided exclusively by the owner's family;
- Interest on working capital: for machinery, equipment and buildings, these have been calculated together with the quotas [28]. The interest on livestock capital was calculated instead by multiplying the average annual interest rate of 4% by the average value of the herd during the year 2022;
- General expenses: they referred to water, electricity, duties and taxes, managerial work and land benefits. In particular, taxes and fees, electricity and water costs were obtained from the information provided directly by the interviewed entrepreneurs. Management expenses were calculated using a percentage of 3% on the TO, net of contributions from the CAP. The land benefit was obtained using the current average annual rent in the areas surveyed for similar funds.

Finally, to evaluate the convenience of the farms under study to activate forms of short supply chains, and in this case, the convenience of transforming the milk produced by building a dairy, the transformation value of the milk was used as a reference parameter, given by the difference between the market value of the goods obtained from the transformation (cheese and ricotta) and the costs incurred to carry out the transformation (fixed and variable costs of the cheese factory).

*2.3. Environmental Analysis*

According to ISO 14040-44 [29,30], the LCA approach was used to estimate the environmental impacts through its four phases: goal and scope definition, life cycle inventory, life cycle impact assessment and interpretation. So, the aim of this analysis was the estimation of the environmental sustainability of three types of livestock farms characteristic of the Cilento area. The reference period of the analysis was set to the end of one production cycle, precisely at the end of 2022. The system boundaries, reported in Figure 1, went from the extraction of raw materials (inputs) to the farm gate (sale of animals) and included all the farm operations characteristics of the different analyzed animal husbandries (the cultivation of the farmland, the grazing of the animals and the management of the stable). All inputs (fuel, lubricants, feed, straw, hay, water, etc.) were included, considering their manufacturing processes. In order to improve the interpretation of environmental results, the whole farm and one ABU were chosen as functional units (FU).

Table 1 reports primary data on the features of the livestock farms collected in situ during the last agricultural year using a data collection sheet. Depending on the analyzed farm, the following farming operations were considered: forage, durum wheat and hay production (soil tillage, other crop-specific operations, harvesting, transport and storage), animal feed, livestock and manure management. The use of primary data (material input types and the amounts used), as in previous studies [31–33], was given priority. Additionally, as a standard practice in LCAs, the active ingredient of each product as well as the amounts of fuel, water and feed consumed were considered for calculation and used in the analysis for each operation to estimate direct and indirect emissions.

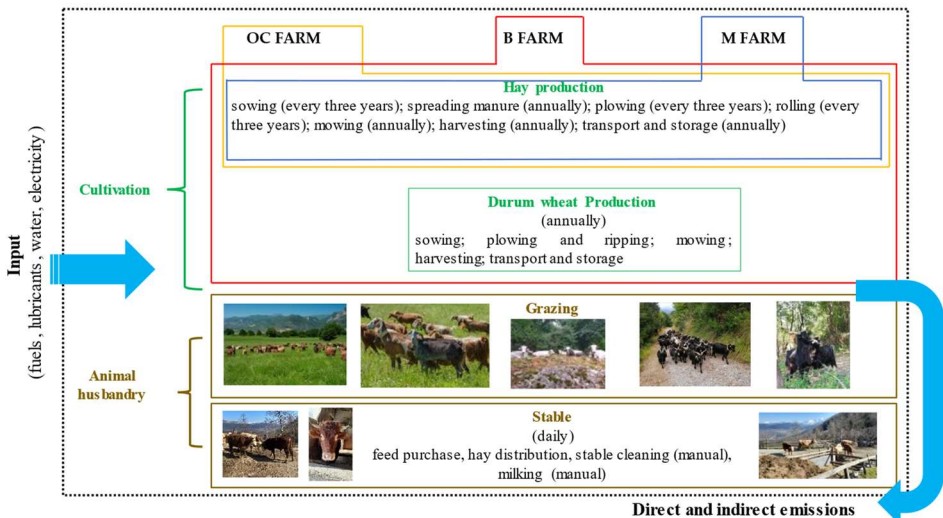

**Figure 1.** The system boundaries used for the environmental and economic analysis (OC FARM: the farm with ovines and caprines; B FARM: the farm with only bovines; M FARM: the farm mixed, namely with cattle, sheep and goats).

**Table 1.** Farm inputs used in the LCA analysis (OC FARM: farm with only sheep and goats; B FARM: farm with only cattle; M FARM: farm with cattle, sheep and goats).

|  | OC FARM | B FARM | M FARM |
|---|---|---|---|
| Human labor (h/year) | 927 | 1267 | 628 |
| Diesel (kg/year) | 1581 | 1870 | 301 |
| Water ($m^3$/year) | 300 | 255 | 195 |
| Feed (kg/year) | 11,700 | 16,200 | 6120 |
| Hay (kg/year) | 36,800 | 40,500 | 24,500 |

The estimation of emissions (direct and embodied) from fuel, lubricants and feed was performed using international databases of scientific importance and reliability, like Ecoinvent 3 [34].

Emissions from enteric fermentation (methane ($CH_4$)) and manure management ($CH_4$, nitrous oxide ($NO_2$), and ammonia ($NH_3$)) by grazing and stabled animals were estimated through the Intergovernmental Panel Climate Change (IPCC) methodology [35,36] and methods and emission factors used are listed in Table 2. As stated in Gavrilova et al. [37], carbon dioxide ($CO_2$) emissions from livestock were not estimated since annual net $CO_2$ emissions are assumed to be zero (the $CO_2$ photosynthesized by plants is returned to the atmosphere as respired $CO_2$) [37].

**Table 2.** Emissions' source, equation and emission factor used.

| Source | Pollutant | Equations | Emission Factor * | Reference |
|---|---|---|---|---|
| Enteric fermentation | $CH_4$ | $CH_4 = N\ heads \cdot EF1$ | EF1 for sheep high productivity systems: 9<br>EF1 for dairy cattle in Western Europe: 126<br>EF1 for other cattle in Western Europe: 52 | Gavrilova et al. [37]—Tier 1 |
| Manure management | $CH_4$ | $CH_4 = [\Sigma(N \cdot VS \cdot AWMS \cdot EF2)/1000]$ | VS for sheep in Western Europe: 8.2<br>VS for goats in Western Europe: 9<br>VS for dairy cattle in Western Europe: 7.5<br>VS for other cattle in Western Europe: 5.7<br>AWMS pasture/range/paddock for sheep (meat) in Western Europe: 0.87<br>AWMS pasture/range/paddock for sheep (dairy) in Western Europe: 0.78<br>AWMS pasture/range/paddock for goats in Western Europe: 0.72<br>AWMS pasture/range/paddock for dairy cattle in Western Europe: 0.26<br>AWMS pasture/range/paddock for not dairy cattle: 0.48<br>EF2 solid storage temperate climate zone for sheep: 5.1<br>EF2 solid storage temperate climate zone for goats: 4.8<br>EF2 solid storage temperate climate zone for dairy cattle: 6.4<br>EF2 solid storage temperate climate zone for non-dairy cattle: 4.8 | Gavrilova et al. [37]—Tier 1 |
| Managed soils | Direct $N_2O$ | $N_2O = [\Sigma\ (F_{ON} \cdot EF3 + F_{PRP} \cdot EF4)] \cdot 44/28$ | EF3: 0.01<br>EF4 for cattle (dairy, non-dairy): 0.02<br>EF4 for sheep and "other animals": 0.01 | IPCC [35]—Tier 1 |
| N volatilized from managed soils | Indirect $N_2O$ | $N_2O_{ind} = [((F_{ON} + F_{PRP}) \cdot Frac_{GASM}) \cdot EF5)] \cdot 44/28$ | EF5: 0.01<br>$Frac_{GASM}$: 0.20 | IPCC [36]—Tier 1 |

* EF1: enteric fermentation emission factors for Tier 1 method—kg $CH_4$/head/yr; VS: annual average excretion per head of species/category, for productivity system in kg VS/animal/yr; AWMS: Animal Waste Management System—regional averages; EF2: methane emission factors by animal category, manure management system and climate zone—g $CH_4$ kg/VS; $F_{ON}$: annual amount of managed animal manure additions applied to soils—kg N/yr; EF3: emission factor for $N_2O$ emissions from N inputs, kg $N_2O$—N/kg N input; $F_{PRP}$: annual amount of urine and dung N deposited by grazing animals on pasture, range and paddock—kg N/yr; EF4: emission factor for $N_2O$ emissions from urine and dung N deposited on pasture, range and paddock by grazing animals, kg $N_2O$-N/kg N input; $Frac_{GASM}$: fraction of applied organic N fertilizer materials ($F_{ON}$) and of urine and dung N deposited by grazing animals ($F_{PRP}$) that volatilizes as $NH_3$ and NOx, kg N volatilized/kg of N applied or deposited; EF5: emission factor for $N_2O$ emissions from atmospheric deposition of N on soils and water surfaces, kg N–$N_2O$/kg $NH_3$–N + NOx–N volatilized).

The impact assessment was performed with the SimaPro 9 software, according to the problem-oriented CML method developed by the Institute of Environmental Sciences of the University of Leiden [38]. The impact categories considered were abiotic depletion (AD), abiotic depletion (fossil fuels) ($AD_{fossil\ fuels}$), global warming potential (GWP) or climate change, photochemical oxidation (PO), ozone layer depletion (OLD), human toxicity (HT), freshwater aquatic ecotoxicity (FEW), marine aquatic ecotoxicity (MAE), terrestrial ecotoxicity (TE), acidification of the air (AA) and eutrophication (EU).

## 3. Results

### 3.1. Farms' Description

The survey showed that two of the farms studied were located in mountains at over 700 m a.s.l. (B FARM and M FARM) and one in a hilly area (OC FARM) at an altitude of about 350 m a.s.l. (Table 3).

**Table 3.** The main characteristics of the analyzed farms (OC FARM: farm with only sheep and goats; B FARM: farm with only cattle; M FARM: farm with cattle, sheep and goats).

|  | OC FARM | B FARM | M FARM |
|---|---|---|---|
| Altitude (m a.s.l.) | 350 | 700 | 700 |
| Farm extension (ha) | 15: 3 owned and 12 rented | 19: 10 owned and 9 rented | 10: 4 owned and 6 rented |
| Technical–economic orientation | Cereal fodder with a zootechnical orientation | Cereal fodder with a zootechnical orientation | Cereal fodder with a zootechnical orientation |
| State-owned pasture (ha) | 15 | 40 | 60 |
| Water sources | 2 farm wells + natural springs | 6 company wells + natural springs + rural aqueduct | 2 company wells + natural springs + rural aqueduct |
| Annual Work Units (AWU) | 3 | 2 | 2 |
| AWU full-time | 1 | 1 | 1 |

In all three cases, the land used was slightly sloping but interspersed with flat spaces. These lands were used almost exclusively (apart from the family gardens) to produce hay consisting of dry polyphyte meadows. In the case of OC FARM, there were also some olive trees (about 20), whose oil was used to meet family needs. B FARM also cultivated two hectares of durum wheat (in rotation), whose grain was used in part to meet the needs of the family and the remaining part as feed for pigs and other farmyard animals, the latter also being intended for self-consumption. Not infrequently, the grain caryopsis was also administered to cattle, like the straw produced.

In all examined farms, the form of management was direct cultivator, in which all the necessary work was provided by the conductor/owner's family; the corporate surface was made up of owned land and rented land; an important role was played by the state-owned lands, the use of which was linked to the payment of the pasture fee. An important element on state-owned lands was the watering points for the animals, which were not always adequate for the needs of grazing animals.

Table 4 shows the characteristics of the analyzed farms, which can be summarized as follows:

- A total of 130 heads were present in OC FARM, of which 110 were adults (40 goats and 70 sheep) and 20 were aged between zero and one year (10 goats and 10 sheep). The sex ratio was one male to about nineteen adult females. The sheep and goats raised were of the mestizo type with an influence of selected breeds (*camosciata delle Alpi* and *Valle del Belice*);

- The herd with only cattle (B FARM) consisted of twenty heads divided as follows: fourteen reproducers, of which there was a 36-month-old bull; two calves aged between 1 and 2 years (one male and one female); four calves less than 1 year old. All the animals were of the mestizo type obtained by crossing the red-spotted breed with *limousines* and local cattle;

- The mixed breeding farm (M FARM) consisted of 64 heads, of which there were 48 goats, 9 sheep and 7 cattle. In particular, for the goats, there were 40 aged over one year and 8 aged less than one year. Among the adult goats, there were two males and the rest were calved goats. The sex ratio was one male to about twenty females. Among the sheep, four were less than one year old and five were adults, including a ram. As for the cattle, there were four cows and three calves. Both the cattle and the sheep and goats were mestizo animals obtained by crossing Alpine and Murcian breeds with local ones.

**Table 4.** The main characteristics of the examined animal husbandry (OC FARM: farm with only sheep and goats; B FARM: farm with only cattle; M FARM: farm with cattle, sheep and goats).

|  | OC FARM | B FARM | M FARM |
|---|---|---|---|
| Breeds | Goats: local mestizos and alpine type; sheep: local mestizos, with rams of the Belice valley breed | Cattle: Pied Red crossbred with limousine male | Sheep and cattle: local mestizos; goats: local mestizos, crossings with subjects of the Murciana breed |
| Number of animals | 130: 80 sheep; 50 goats | 20 cattle | 64: 9 sheep; 48 goats; 7 cattle |
| ABU * | 20 | 17 | 13 |
| Farming system | Semi-extensive | Semi-extensive | Semi-extensive |
| Replacement | Internal | Internal and external | Internal |

* Adult bovine unit.

In all analyzed cases, the rearing system was semi-extensive, in which the animals always lived on pastures during the day, with the exception of cold or very rainy winter days during which they were kept in stables and fed with hay and mixtures based on bran, cereals and legumes prepared by the farmers themselves. In the winter period, it was widespread practice to feed the animals with the mixtures even on days when the animals were grazing. In a few cases, the diet was integrated with foods external to the farm, but in any case, it was simple feed.

The weaning of young animals was carried out with mother's milk and lasted about three months in the case of calves and 40–50 days in the case of lambs and kids destined for slaughter. The same was extended to about 6 months in the case of replacement animals. The latter was always internal, except for the occasional purchase of male reproducers from outside.

### 3.2. Profitability Analysis of Studied Livestock Farms

Table 5 shows the results of the economic analysis both *per* farm and per ABU. With reference to the total output, the latter assumed higher values in breeding with only cattle (about 35,000 EUR/farm/year and 2049 EUR/ABU), followed by mixed breeding (more than 28,000 EUR/farm/year and 2183 EUR/ABU) and finally sheep–goat farming (about 16,300 EUR/farm/year and 813 EUR/ABU). At the same time, in farms B and M, more than 70% of the TO was derived from CAP aids, while in farm OC, it was mostly derived from the sale of products (lambs and kids) (57%).

Referring to the various items that make up the costs, Table 5 shows that the highest costs were recorded in the farm with cattle breeding, where over 29,000 EUR/farm/year have been calculated, while the lowest costs were found in the sheep and goat farming (almost 16,000 EUR/farm/year). In all the farms analyzed, fixed costs exceeded 60% of the total costs and even 70% in the mixed one. Among these, an important role was played by quotas and interests, in which M FARM represents 43% of the total costs. The reason must essentially be sought in the fact that the costs relating to the machinery and equipment were calculated considering their new and current value, excluding the possible forms of incentives from which, on the contrary, the interviewed entrepreneurs benefited at the time of purchase, but at the moment of the interview, they did not remember.

Among the various components of variable costs, an important role was played by the cost of feeding livestock, which accounted for 13% of the total costs in mixed breeding, 14% in sheep and goat breeding and 25% in breeding with only cattle. The reasons for the significant incidence of these costs in specialized cattle breeding were due to the composition of the particularly rich food ration and the low quantity of farm forage produced in 2022, the latter caused by the damage that the pastures and meadows suffered in the presence of wild boars.

**Table 5.** Results of the economic analysis *per* farm and *per* ABU. Annual values. (OC FARM: farm with only sheep and goats; B FARM: farm with only cattle; M FARM: farm with cattle, sheep and goats.)

| | OC FARM | | | B FARM | | | M FARM | | |
|---|---|---|---|---|---|---|---|---|---|
| | EUR *per* Farm | EUR *per* ABU * | % | EUR *per* Farm | EUR *per* ABU* | % | EUR *per* Farm | EUR *per* ABU * | % |
| Products sold | 9300 | 465 | 57% | 7400 | 435 | 21% | 5550 | 427 | 20% |
| Gross profit of the stable | 0 | 0 | 0% | 1480 | 87 | 4% | 1200 | 92 | 4% |
| CAP contributions | 6965 | 348 | 43% | 25,952 | 1527 | 75% | 21,630 | 1664 | 76% |
| **Total Output** | **16,265** | **813** | **100%** | **34,832** | **2049** | **100%** | **28,380** | **2183** | **100%** |
| Livestock feed expenses | 2200 | 110 | 14% | 7220 | 425 | 25% | 3290 | 253 | 13% |
| Veterinary and health expenses | 1800 | 90 | 11% | 1300 | 76 | 4% | 1150 | 88 | 5% |
| Costs for forage farming | 955 | 48 | 6% | 2640 | 155 | 9% | 3000 | 231 | 12% |
| **Variable costs** | **4955** | **248** | **31%** | **11,160** | **656** | **38%** | **7440** | **572** | **29%** |
| Quotas and interests (machinery, equipment and property) | 4038 | 202 | 26% | 10,561 | 621 | 36% | 10,964 | 843 | 43% |
| Livestock capital interest | 1520 | 76 | 10% | 1368 | 80 | 5% | 730 | 56 | 3% |
| Rent and grazing fee | 750 | 38 | 5% | 1800 | 106 | 6% | 1640 | 126 | 7% |
| Overheads (management+taxes+other expenses) | 4529 | 226 | 29% | 4516 | 266 | 15% | 4453 | 343 | 18% |
| **Fixed costs** | **10,837** | **542** | **69%** | **18,245** | **1073** | **62%** | **17,787** | **1368** | **71%** |
| **Total costs** | **15,792** | **790** | **100%** | **29,405** | **1730** | **100%** | **25,227** | **1941** | **100%** |
| **Gross profit (TO-VC)** | **11,310** | **566** | | **23,672** | **1392** | | **20,940** | **1611** | |
| **Net income (TO-TC)** | **473** | **24** | | **5427** | **319** | | **3154** | **243** | |

\* Adult bovine unit.

The highest gross profit was recorded on the farm with cattle breeding and was equal to just over EUR 23,600 (the farm that had the highest costs but also the highest TO), followed by M FARM and finally by OC FARM. Also, with reference to net income, farm B was the most profitable, followed by the farm with mixed breeding and finally the farm with sheep and goat breeding (Table 5).

The same cannot be said per ABU: the largest GP was registered in M FARM, followed by B FARM and finally OC FARM (Table 5).

If the CAP contributions were not considered in the TO, the GP would be positive only on the farm with sheep and goat breeding, in which the sale of breeding products covered the variable costs. Referring to net income, if the CAP aids were not considered, all the farms analyzed would be negative, especially the farm with cattle breeding (Table 6).

*3.3. Economic Evaluation of Milk Processing*

Starting from a concrete case comparable to the realities analyzed, Table 7 shows the results of the analysis of the convenience of transforming the milk on the farm, which was currently self-consumed, given away, and only partially transformed.

**Table 6.** Annual economic results *per* farm and *per* ABU without CAP contributions (OC FARM: farm with only sheep and goats; B FARM: farm with only cattle; M FARM: farm with cattle, sheep and goats).

|  | OC FARM | | B FARM | | M FARM | |
|---|---|---|---|---|---|---|
|  | **EUR *per* Farm** | **EUR *per* ABU \*** | **EUR *per* Farm** | **EUR *per* ABU \*** | **EUR *per* Farm** | **EUR *per* ABU \*** |
| Total output (TO) | 9300 | 465 | 8880 | 522 | 6750 | 519 |
| Variable costs | 4955 | 248 | 11,160 | 656 | 7440 | 572 |
| Fixed costs | 10,837 | 542 | 18,245 | 1073 | 17,787 | 1368 |
| Total costs | 15,792 | 790 | 29,405 | 1730 | 25,227 | 1941 |
| Gross profit (TO-CV) | 4345 | 217 | −2280 | −134 | −690 | −53 |
| Net income (TO-CT) | −6492 | −325 | −20,525 | −1207 | −18,477 | −1421 |

\* Adult bovine unit.

**Table 7.** Cost of milk processing and value of processing. Annual values.

|  | kg | EUR |
|---|---|---|
| Milk processed in 2022 | 12,600 |  |
| Values of products sold (cheese and ricotta) |  | 18,600 |
| Dairy depreciation quota |  | 819 |
| Equipment depreciation quota |  | 462 |
| Dairy maintenance and insurance quota |  | 1600 |
| Equipment maintenance and insurance quota |  | 880 |
| General management expense |  | 6100 |
| Management |  | 558 |
| Labor (implicit cost) |  | 0 |
| Total costs |  | 10,419 |
| Net income |  | 8181 |
| Value of processed products per kg of milk |  | 1.48 |
| Cost of processing per kg of milk |  | 0.83 |

The objective of this analysis was to evaluate the convenience of the possible inclusion of a mini-dairy on the three farms examined and to verify whether the decision to move towards the transformation of milk and the production of dairy products was a convenient choice from an economic point of view. To this end, starting from a real case, the construction of a mini-dairy ex novo and the use of family labor were hypothesized. Under these conditions, the interview carried out showed that the value of the processed products amounted to approximately EUR 18,600, which was equal to EUR 1.48 per liter of processed milk, much higher than the market price of raw milk recorded on the local market (sheep's milk: 1.00–1.20; goat's milk: 0.80–1.20; cow's milk: data not available). At the same time, the costs amounted to almost EUR 10,500, i.e., EUR 0.83 per liter of processed milk. Consequently, the transformation of the milk would ensure the farmer an additional net income of more than EUR 8000 (Table 7). Conversely, the inclusion of a work unit working 4 h/day for 280 days/year would entail an additional labor cost of around 8400 EUR/year and consequently a negative net income.

### 3.4. Results of the Environmental Analysis

The results of the environmental analysis are shown in Table 8, which highlights that, at the farm level, the mixed farm (M FARM) was the most sustainable, showing lower impacts than the other two analyzed farms with reference to all impact categories considered. In particular, its management annually causes the consumption of nearly 44,000 MJ of energy of fossil origin; the emission of 47,659 kg of $CO_2$ eq, of more than 3,330,000 kg of 1,4 dichlorobenzene eq causing human, water and terrestrial ecotoxicity; of 451 kg of $SO_2$ eq causing air acidification; and 165 kg of phosphates causing eutrophication. Conversely, B FARM with only cattle was found to have the most impact (Table 8).

**Table 8.** Results of the environmental analysis. Annual values, *per* farm, broken down by emission source and impact category (AD—abiotic depletion; AD$_{fossil\ fuels}$—abiotic depletion (fossil fuels); GWP—global warming potential; OLP—ozone layer depletion; HT—human toxicity; FEW—freshwater aquatic ecotoxicity; MAE—marine aquatic ecotoxicity; TE—terrestrial ecotoxicity; PO—photochemical oxidation; AA—air acidification; EU—eutrophication) (OC FARM: farm with only sheep and goats; B FARM: farm with only cattle; M FARM: farm with cattle, sheep and goats).

| Impact Category | Unit | Total | Farm Operations | Animal Feed | Enteric Fermentation | Manure Management | Nitrogen Emissions |
|---|---|---|---|---|---|---|---|
| | | | | **OC FARM** | | | |
| AD | kg Sb eq | 0 | 0.1 | 0.0 | 0.0 | 0.0 | 0.0 |
| AD$_{fossil\ fuels}$ | MJ | 156,798 | 114,646.8 | 42,150.8 | 0.0 | 0.0 | 0.0 |
| GWP | kg $CO_2$ eq | 57,111 | 9078.2 | 5850.0 | 32,760.0 | 2268.0 | 7155.0 |
| OLP | kg CFC-11 eq | 0 | 0.0 | 0.0 | 0.0 | 0.0 | 0.0 |
| HT | kg 1.4-DB eq | 8580 | 5566.4 | 2991.2 | 0.0 | 0.0 | 22.8 |
| FEW | kg 1.4-DB eq | 6831 | 4376.5 | 2454.9 | 0.0 | 0.0 | 0.0 |
| MAE | kg 1.4-DB eq | 11,474,436 | 8,025,201.0 | 3,449,235.2 | 0.0 | 0.0 | 0.0 |
| TE | kg 1.4-DB eq | 321 | 28.0 | 292.7 | 0.0 | 0.0 | 0.0 |
| PO | kg $C_2H_4$ eq | 11 | 2.2 | 0.9 | 7.0 | 0.5 | 0.0 |
| AA | kg $SO_2$ eq | 580 | 56.1 | 158.9 | 0.0 | 0.0 | 364.8 |
| EU | kg $PO_4$—eq | 251 | 16.1 | 147.4 | 0.0 | 0.0 | 87.1 |
| | | | | **B FARM** | | | |
| AD | kg Sb eq | 0 | 0.1 | 0.1 | 0.0 | 0.0 | 0.0 |
| AD$_{fossil\ fuels}$ | MJ | 193,925 | 135,562.5 | 58,362.7 | 0.0 | 0.0 | 0.0 |
| GWP | kg $CO_2$ eq | 81,275 | 10,734.3 | 8100.0 | 50,232.0 | 1344.0 | 10,865.0 |
| OLP | kg CFC-11 eq | 0 | 0.0 | 0.0 | 0.0 | 0.0 | 0.0 |
| HT | kg 1.4-DB eq | 10,759 | 6582.0 | 4141.6 | 0.0 | 0.0 | 35.1 |
| FEW | kg 1.4-DB eq | 8574 | 5174.9 | 3399.1 | 0.0 | 0.0 | 0.0 |
| MAE | kg 1.4-DB eq | 14,265,147 | 9,489,282.5 | 4,775,864.1 | 0.0 | 0.0 | 0.0 |
| TE | kg 1.4-DB eq | 438 | 33.1 | 405.3 | 0.0 | 0.0 | 0.0 |
| PO | kg $C_2H_4$ eq | 15 | 2.6 | 1.2 | 10.8 | 0.3 | 0.0 |
| AA | kg $SO_2$ eq | 848 | 66.3 | 220.0 | 0.0 | 0.0 | 561.6 |
| EU | kg $PO_4$—eq | 357 | 19.0 | 204.1 | 0.0 | 0.0 | 133.9 |
| | | | | **M FARM** | | | |
| AD | kg Sb eq | 0 | 0.0 | 0.0 | 0.0 | 0.0 | 0.0 |
| AD$_{fossil\ fuels}$ | MJ | 43,885 | 21,837.0 | 22,048.1 | 0.0 | 0.0 | 0.0 |
| GWP | kg $CO_2$ eq | 47,659 | 1729.1 | 3060.0 | 34,608.0 | 1372.0 | 6890.0 |
| OLP | kg CFC-11 eq | 0 | 0.0 | 0.0 | 0.0 | 0.0 | 0.0 |
| HT | kg 1.4-DB eq | 2647 | 1060.2 | 1564.6 | 0.0 | 0.0 | 22.3 |
| FEW | kg 1.4-DB eq | 2118 | 833.6 | 1284.1 | 0.0 | 0.0 | 0.0 |
| MAE | kg 1.4-DB eq | 3,332,788 | 1,528,573.1 | 1,804,215.3 | 0.0 | 0.0 | 0.0 |
| TE | kg 1.4-DB eq | 158 | 5.3 | 153.1 | 0.0 | 0.0 | 0.0 |
| PO | kg $C_2H_4$ eq | 9 | 0.4 | 0.5 | 7.4 | 0.3 | 0.0 |
| AA | kg $SO_2$ eq | 451 | 10.7 | 83.1 | 0.0 | 0.0 | 356.8 |
| EU | kg PO—eq | 165 | 3.1 | 77.1 | 0.0 | 0.0 | 85.1 |

In the three analyzed farms, the percentage contribution of the various sources of emissions to the different impact categories, shown in Figure 2, highlights that the farm operations (production of hay, forage and durum wheat) had a significant impact on the consumption of abiotic resources, on ozone layer depletion and human, terrestrial and aquatic toxicity; enteric fermentation was mainly the cause of global warming and photochemical oxidation, especially in farm M, where it represented 73% of $CO_2$ eq emissions and 86% of ethylene emissions; animal feed accounted for more than 90% of terrestrial ecotoxicity and nitrogen emissions mainly causing air acidification and eutrophication (Figure 2). These last results agree with Bragaglio et al. [39] and Castanheira et al. [40], who found that the largest source of emissions to air and water from cattle production

systems was the production of concentrates, even if, as in our case, it was not a question of real concentrated feed but of mixtures and simple foods. On the contrary, manure management contributed slightly to the impacts and only on photochemical oxidation and global warming, with percentages ranging from 2% (in B FARM) to 5% (in OC FARM) and from 2% (in B FARM) to 4% (in OC FARM), respectively. Similar values were found by Bragaglio et al. [39].

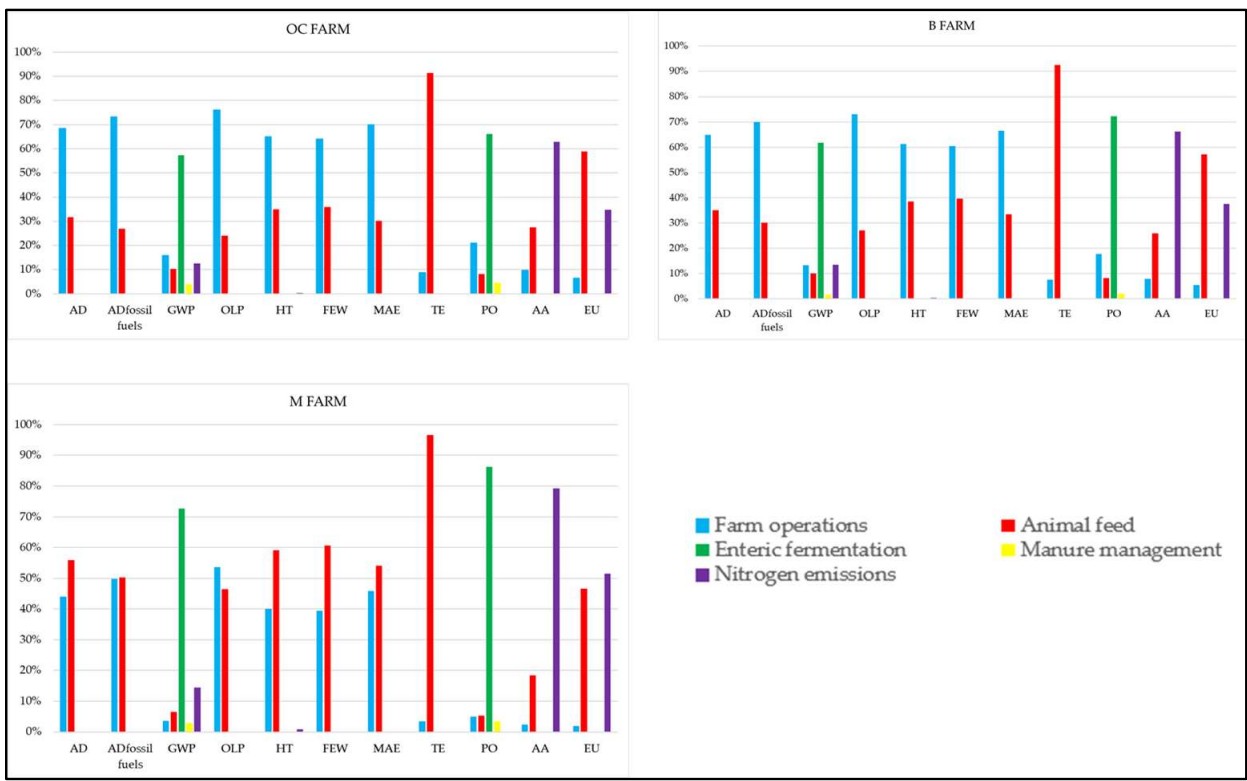

**Figure 2.** Percentage contribution of the various emission sources to the different impact categories in the analyzed farms (AD—abiotic depletion; AD$_{fossil fuels}$—abiotic depletion (fossil fuels); GWP—global warming potential; OLP—ozone layer depletion; HT—human toxicity; FEW—freshwater aquatic ecotoxicity; MAE—marine aquatic ecotoxicity; TE—terrestrial ecotoxicity; PO—photochemical oxidation; AA—air acidification; EU—eutrophication) (OC FARM: farm with only sheep and goats; B FARM: farm with only cattle; M FARM: farm with cattle, sheep and goats).

Referring to global warming, one of the issues most felt by the community, data *per* ABU partially confirm data *per* farm: B FARM remained the one with the most impact, while the farm with sheep and goats (OC FARM) was the most sustainable. The same figure shows that in all three farms, the major impacts were due to enteric fermentation, which represented 57%, 62% and 73% of the total impacts in OC FARM, B FARM and M FARM, respectively (Figure 3). Similar results about the fundamental role played by enteric fermentation on GWP, especially in cattle farms, were obtained by Bragaglio et al. [39], Dick et al. [41] and Ruviaro et al. [42].

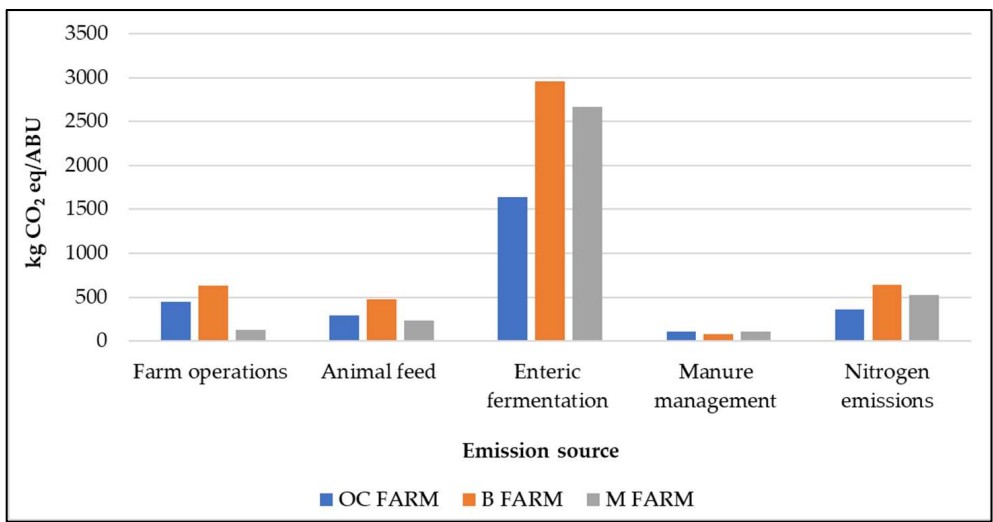

**Figure 3.** Global warming *per* ABU (adult bovine unit) distinguished by emission sources. (OC FARM: farm with only sheep and goats; B FARM: farm with only cattle; M FARM: farm with cattle, sheep, and goats).

## 4. Discussion

From the interviews carried out, important considerations emerge that can be extended to other farms located in inland areas. In particular, it emerged that the farms analyzed had many aspects in common: the form of management (direct farm); the organization of the work (purely family-run); the use of the utilized agricultural area (UAA); the title of ownership of the land (partly owned and partly rented); the technical–productive organization (zootechnical type); the semi-extensive breeding system with a broad use of pasture; 90% of the forage produced on the farm, which was used in the form of hay and was administered only when the pasture was not sufficient to cover the needs of the animals. Furthermore, in all three farms analyzed, the production orientation was that of meat, even if modest quantities of milk were transformed, especially in spring, whose transformed products were destined for gifts and self-consumption. The main method of marketing animals was the sale to wholesalers (stall and slaughter cattle), while for sheep and goats, in addition to wholesalers, there was also the sale to local retailers (butchers). At the same time, the farms differed from each other, especially with reference to agricultural and land capital: minimal in sheep and goat breeding, while more significant in the other two.

The economic analysis showed that the survival of farms in inland areas is essentially linked to public subsidies, which in some cases represent more than 75% of the total output. This situation is currently a source of destabilization for the farmers, who are aware that the future of their breeding activities is strongly conditioned by public incentives, all aggravated by the increase in management costs, in particular those for feeding livestock, which aggravate an already considerably compromised reference framework. In fact, there are several issues that concern farmers. Some are related to the market of both products and production factors. Referring to products, the concerns are linked to the crisis that has affected the kid and lamb market in the last two years (despite the increase in production costs, producer prices have practically remained unchanged). A no less serious problem is the market for stalled cattle, which has seen a constantly fluctuating sale price due to the fickleness of the meat market. Further problems concern the allocation of state land, which is subject in many cases to restrictions due to extreme causes, such as natural disasters, and the lack of attention from local authorities in reference to water supply points during the summer and in the pastures, which are considerably scarce. Moreover, the difficulties created by wildlife are growing, in particular by wild boars, which damage meadows and pastures, and by feral wolves and dogs, which attack grazing animals, especially young

ones. Consequently, it would be necessary to build adequate fences to protect animals and crops, but since these are protected areas, this goes against the Park's policy, which does not allow for fences.

The analysis of the convenience of processing the milk on the farm showed that the convenience (given by a further positive net income) could only be achieved if family labor was employed, and therefore the cost of labor was an implicit cost. At the same time, from the comparisons with the farmers interviewed, it emerged that:

- OC FARM currently has family labor to be able to engage in the transformation of the milk but lacks a building area to build the dairy and the financial resources necessary for the construction;
- B FARM has both family workers to use in the cheese factory and owns a room that could be renovated and used for cheese making in order to further reduce costs;
- M FARM, on the other hand, did not show any interest in the higher margin that can be collected from the processing of the milk.

From what has been said, it can be deduced that the concept of family enterprise plays a fundamental role in management decisions, in the size of animal breeding, and in investment decisions. Entrepreneurial choices depend on a delicate balance between market demands, domestic strategies to maintain or achieve the right family size and composition and the availability of local resources. The family maintains a central economic and productive role and is instrumental in keeping pastoral agriculture alive. Some breeders today belong to families that have dedicated themselves uninterruptedly to sheep farming for some time [43,44].

Referring to environmental impacts, the analysis showed the lower sustainability of cattle farming, mainly due to the higher methane emissions during enteric fermentation. At the same time, it should be noted that the results obtained in the present research, although elaborated *per* farm and *per* ABU, were considerably lower than those of other studies, especially concerning bovine farming [38,41,42,45,46]. The decision to use the entire farm as FU and not the kg of meat or milk, as is usually performed in LCA studies concerning livestock systems [9], was linked to the fact that livestock farms often produce multiple products [47,48], such as milk and/or meat, but also cereals, hay and several by-products (e.g., manure, skin, wool and straw) [45]. So, similarly to the economic analysis, which analyzed the results of the farms as a whole, the environmental analysis wanted to estimate the impacts as a whole. Then, through an appropriate allocation (usually of an economic nature) [45], the impacts would be distributed among the different products and services offered, a step that will be the subject of future research.

In addition to considering the products that different farms produce, the ecosystem services they offer should also be considered. In fact, the public goods that grazing farms offer are remarkable, especially in inland and marginal areas: landscape conservation [49,50], enhancement of biodiversity [51] and wildfire prevention [52,53]. Moreover, as stated by Hoffmann et al. [5], in addition to the already known provisioning services (products obtained), several are the benefits obtained from the regulation of ecosystem processes (waste recycling and conversion of non-human edible feed; land degradation and erosion prevention; water quality regulation; regulation of water flows; pollination; biological control and animal/human disease regulation) and supporting services, namely ecosystem services that are necessary for the production of all other ecosystem services. The latter, in the case of grazing animals, refers to the maintenance of soil structure and fertility; improvement in vegetation growth and cover; maintenance of life cycles of species; habitat connectivity through seed dispersal in guts and manure; and maintenance of genetic diversity. Alongside these important ecosystem services, extensive animal husbandry in inland areas such as the one covered by this study also offers cultural services: it helps to maintain elements of the local culture that are valued as part of the heritage of the region (cultural identity), it becomes part of the landscape shaped by the animals themselves, and it is often part of religious rites and ceremonies [5,45].

Therefore, analyses of economic and environmental sustainability should take all of this into account in order to reward farmers who choose to stay in inland areas, which are very often marginal, and to encourage them to continue their farming activity for all the ESs they offer, even if it is not very profitable.

## 5. Conclusions

From the analysis of the three farms with animal breeding located in inland areas of the National Park of Cilento, Vallo di Diano and Alburni, it emerged that the family enterprise conditions the entrepreneurial choices and the dimension/type of breeding, and that cattle breeding was the most profitable but also the most impactful. At the same time, cattle breeding's higher net income was mainly due to the CAP contributions it received, and therefore its future depends on them.

The impacts calculated here with the LCA methodology were not allocated among the many products and services offered by the three farms analyzed, as the farms in their entirety and the ABU were deliberately chosen as FU in order to better compare the farms subject to the study. Furthermore, the estimation of the many ecosystem services offered by grazing animals is not always easily quantifiable but will be the subject of future research.

**Author Contributions:** Conceptualization, M.C. and M.P.; methodology, M.C. and M.P.; validation, M.C. and M.P.; formal analysis, M.P.; investigation, A.I. and F.D.B.; resources, M.C.; data curation, M.P.; writing—original draft preparation, M.P.; writing—review and editing, M.C. and M.P.; visualization, M.C., F.D.B. and A.I.; supervision, M.C.; project administration, M.C.; funding acquisition, M.C. All authors have read and agreed to the published version of the manuscript.

**Funding:** This research was partially funded by "PSR Campania Region 2014–2020. Nobili Cilentani Project: Applicazione del metodo nobile ad alcune produzioni zootecniche cilentane". Unique Project Code: E66B19000630009.

**Institutional Review Board Statement:** Not applicable.

**Informed Consent Statement:** Informed consent was obtained from all subjects involved in the study.

**Data Availability Statement:** Not applicable.

**Acknowledgments:** We are grateful to the interviewed breeders for giving us access to their field data.

**Conflicts of Interest:** The authors declare no conflict of interest.

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
