# Peer review of "Evaluation of the Economic and Environmental Sustainability of Livestock Farms in Inland Areas"

_agriculture, doi:10.3390/agriculture13091708_

Round 1

Reviewer 1 Report

General comment

The study by Cerrato et al. examined the economic and environmental sustainability of three types of livestock farms situated in the inland areas of southern Italy. The current manuscript appears to be based on their previous study, where the authors highlighted concerns of the farmers regarding the economic viability of these farms. The manuscript was engaging, easy to follow, and well organized. Please take into consideration the following suggestions to further enhance the quality of the manuscript.  

Abstract

Line 10: The word “evaluation” itself constitutes a scientific contribution. Therefore, explicitly stating the aim of the study was to provide a scientific contribution might be of lesser significance.

Line 11-12: The study location should be described as "inland areas of southern Italy" instead of mentioning the names of the three different regions.

Introduction

The introduction effectively highlights the importance of livestock and its ecological role, underscoring the study's relevance for the inland areas of southern Italy. Please, consider the following suggestions to improve the introduction:

Line: 54-56: Consider revising the sentence for improved clarity and reader comprehension.

Line 81-99: Rather than providing a list of specific study titles, the authors should provide a brief and concise overview of recent Life Cycle Assessment (LCA) studies conducted. To achieve this, the authors could categorize these studies, for instance, mentioning that several studies focusing on small ruminants (reference), bovine (reference), swine (reference), and poultry (reference) have been published.

Line 102-103: See the comment for line 10 in the abstract section.

Material and Methods

Line 111: Please specify that these areas are in Italy, as not everyone may be familiar with their location. Also, add the latitude and longitudes of the area.

Line 127-134: The sentences are understandable, but it might benefit from some slight rephrasing to enhance clarity. A more straightforward wording might make it easier for a broader audience to understand.

Figure 1 was quite easy to understand. Adding a description of the farms in the figure caption would provide helpful context.

Line 241: Please add a reference for this statemen.

Table 4: Provide the description of the abbreviation (ABU) in the footnote to facilitate reader understanding. Please apply this suggestion to all the tables.  Results and Discussion

The results were well explained and presented in a comprehensible manner. The discussion was supported by prior studies and suggested avenues for future research.

The conclusion was concise and aligned with the study objectives.

The language was mostly easy to understand, but there were instances of complex sentences that made comprehension difficult. Additionally, a few minor spelling errors were present. To improve readability, it's suggested to break down longer sentences, particularly those with intricate punctuation, into shorter ones.

Author Response

General comment

The study by Cerrato et al. examined the economic and environmental sustainability of three types of livestock farms situated in the inland areas of southern Italy. The current manuscript appears to be based on their previous study, where the authors highlighted concerns of the farmers regarding the economic viability of these farms. The manuscript was engaging, easy to follow, and well organized. Please take into consideration the following suggestions to further enhance the quality of the manuscript.  

Abstract

Line 10: The word “evaluation” itself constitutes a scientific contribution. Therefore, explicitly stating the aim of the study was to provide a scientific contribution might be of lesser significance.

Thanks for your suggestion. We modified it. Please, see the revised manuscript.

Line 11-12: The study location should be described as "inland areas of southern Italy" instead of mentioning the names of the three different regions.

Thanks for your suggestion. We added it, but the study location is into a National Park situated in one region. Please, see the revised manuscript.

Introduction

The introduction effectively highlights the importance of livestock and its ecological role, underscoring the study's relevance for the inland areas of southern Italy. Please, consider the following suggestions to improve the introduction:

Line: 54-56: Consider revising the sentence for improved clarity and reader comprehension.

Thanks for your suggestion, it's a (poorly written) repetition of what's written further on, so we deleted it. Please, see the revised manuscript.

Line 81-99: Rather than providing a list of specific study titles, the authors should provide a brief and concise overview of recent Life Cycle Assessment (LCA) studies conducted. To achieve this, the authors could categorize these studies, for instance, mentioning that several studies focusing on small ruminants (reference), bovine (reference), swine (reference), and poultry (reference) have been published.

Thanks a lot for your suggestion. We improved it. Please, see the revised manuscript.

Line 102-103: See the comment for line 10 in the abstract section.

Done. Thanks.

Material and Methods

Line 111: Please specify that these areas are in Italy, as not everyone may be familiar with their location. Also, add the latitude and longitudes of the area.

Thanks for your suggestion. We added the reference “Italy”, but, for the purposes of the study, it does not seem necessary to add the latitude and longitudes of the area. But if you think it’s strictly important, we’ll add it.

Line 127-134: The sentences are understandable, but it might benefit from some slight rephrasing to enhance clarity. A more straightforward wording might make it easier for a broader audience to understand.

Thanks a lot for your suggestion. We rewrote these sentences. Please, see the revised manuscript.

Figure 1 was quite easy to understand. Adding a description of the farms in the figure caption would provide helpful context.

Done. Thanks.

Line 241: Please add a reference for this statemen.

Done. Thanks.

Table 4: Provide the description of the abbreviation (ABU) in the footnote to facilitate reader understanding. Please apply this suggestion to all the tables.  

Done. Thanks.

Results and Discussion

The results were well explained and presented in a comprehensible manner. The discussion was supported by prior studies and suggested avenues for future research.

The conclusion was concise and aligned with the study objectives.

Thanks a lot for your comments.

Comments on the Quality of English Language

The language was mostly easy to understand, but there were instances of complex sentences that made comprehension difficult. Additionally, a few minor spelling errors were present. To improve readability, it's suggested to break down longer sentences, particularly those with intricate punctuation, into shorter ones.

Thanks for your suggestions. We improved it. Please, see the revised manuscript.

Reviewer 2 Report

Comment #1

What do you mean by “above all herds” on line 37?

Comment #2

Reword the sentence on lines 138-141.  It is not clear what you are trying to communicate here.  

Comment #3

On line 188 you mention “land benefits”.  Are you referring to cash rent?  Do you include both actual cash rent paid and imputed cash rent from land that is owned?

Comment #4

In Table 3, what do you mean by “economic technical order” and what does “nical” stand for?

Comment #5

In Table 5 and Table 6, “CV” and “CT” when referring to gross profit and net income should be “VC” and “TC”.

My suggestions are included in my comments.

Author Response

Comment #1

What do you mean by “above all herds” on line 37?

We meant cattle breeding. Thanks for your observation. We replaced it with cattle. Please, see the revised manuscript.

Comment #2

Reword the sentence on lines 138-141.  It is not clear what you are trying to communicate here.  

Thanks for your suggestion. We reword the sentence. Please, see the revised manuscript.

Comment #3

On line 188 you mention “land benefits”.  Are you referring to cash rent?  Do you include both actual cash rent paid and imputed cash rent from land that is owned?

Thanks for your observation. With “land benefit” we mean the income generated by the property capital, which is equal to the landed income of a fund net of the taxes imposed on it. It is enjoyed by the owner of the bare land and can be obtained from the rent or through the farm balance sheet, in this case by the current average annual rent in the areas surveyed for similar funds.

Comment #4

In Table 3, what do you mean by “economic technical order” and what does “nical” stand for?

I’m sorry but there were some mistakes and thank you for pointing them out. Anyway, the technical-economic orientation (and not order, sorry) of an agricultural and zootechnical farm expresses the type of the activity that characterizes the farm’s activity. It is determined by the percentage incidence of the standard production of the various production activities of the farm (crops and livestock) with respect to its total standard production. “nical” is the final part of zootechnical. We have corrected these mistakes. Please, see the revised manuscript.

Comment #5

In Table 5 and Table 6, “CV” and “CT” when referring to gross profit and net income should be “VC” and “TC”.

Thank you very much. We corrected these mistakes. Please, see the revised manuscript.

Comments on the Quality of English Language

My suggestions are included in my comments.

Many thanks for your suggestions and revisions. We appreciated them so much.